# Double Duty: Mitotic Kinesins and Their Post-Mitotic Functions in Neurons

**DOI:** 10.3390/cells10010136

**Published:** 2021-01-12

**Authors:** Nadine F. Joseph, Supriya Swarnkar, Sathyanarayanan V Puthanveettil

**Affiliations:** 1The Skaggs Graduate School of Chemical and Biological Sciences, Scripps Research Institute, La Jolla, CA 92037, USA; NJoseph@scripps.edu; 2Department of Neuroscience, The Scripps Research Institute, Jupiter, FL 33458, USA; sswarnka@scripps.edu

**Keywords:** kinesins, synaptic plasticity, structural plasticity, mitosis, transport, microtubule regulation, cancer

## Abstract

Neurons, regarded as post-mitotic cells, are characterized by their extensive dendritic and axonal arborization. This unique architecture imposes challenges to how to supply materials required at distal neuronal components. Kinesins are molecular motor proteins that mediate the active delivery of cellular materials along the microtubule cytoskeleton for facilitating the local biochemical and structural changes at the synapse. Recent studies have made intriguing observations that some kinesins that function during neuronal mitosis also have a critical role in post-mitotic neurons. However, we know very little about the function and regulation of such kinesins. Here, we summarize the known cellular and biochemical functions of mitotic kinesins in post-mitotic neurons.

## 1. Introduction

Neurons have a highly polarized structure that includes axons and dendrites. These spatially distant structures are highly dynamic, metabolically active, and require tight control to maintain normal cell function and morphology [1]. To maintain this control, a system of molecular motor proteins (kinesins, dyneins and myosins) is responsible for the trafficking of various cargos such as receptor subunits, organelles, proteins, and RNAs along the cytoskeleton [1,2]. The kinesin superfamily proteins (Kifs) facilitate the transport of organelles such as mitochondria [3], proteins such as glutamate receptors [4], and RNAs such as CaMKIIα [5] from the cell body to the synapses along microtubule cytoskeleton. Kinesins were discovered in 1985 by Ronald Vale [6] and Scott Brady [7] in the large axons of squids. They found that kinesins were distinct from myosin and dynein in both molecular weight and movement within cells. Since then, research into kinesins has blossomed, with phylogenetic analysis revealing that kinesins in mammals form a large and complex superfamily with 14 subfamilies and 45 members [8]. Most kinesins have a general structure of the motor domain, stalk, and a tail [1]. 

Kinesins are expressed throughout the body but are enriched in the brain. Kinesins can be separated into three different groups, based on the location of the motor domain: N-terminal, C-terminal, and internal (M) kinesins [9]. N-terminal kinesins, N-KIFs, tend to move towards the polymerizing or plus end of the microtubule, whereas, C-KIFs tend to move toward the negative ends of the microtubule [1]. M-type kinesins mainly function at the ends of microtubules to induce depolymerization [9]. Kinesins walk along microtubules unidirectionally using a ‘hand over hand’ mechanism to shuttle cargos throughout the cell. Some Kifs (M-type) are also known to destabilize microtubules. With these two functions, kinesins have been implicated in vesicular and organelle transport, spindle and chromosome motility, and flagellar and ciliary function [10]. 

Neurons are among a few terminally differentiated post-mitotic cells that remain functional for decades. One way for neurons to accommodate this state is by recycling mitotic factors for post-mitotic use [11]. Mitotic kinesins are one such example. Below, we discuss several mitotic kinesins that have post-mitotic function in neurons (Figure 1).

## 2. Mitotic Functions of Kinesins

Mitosis has four consecutive phases; prophase, metaphase, anaphase, and telophase [13]. Kinesins play a vital role in many of the phases of mitosis through their canonical function of transport as well as microtubule regulation. 

Previous research has demonstrated that Kinesin-2 family members participate in organelle transport during interphase [14]. Haraguchi et al. showed that KIF3A/KIF3B (members of the Kinesin-2 family) localizes to the centrosomes in interphase as well as prometaphase [15]. Once the chromosomes condense during prometaphase, KIF3A associates with the newly formed spindle fibers. For the remainder of the phases, KIF3A accumulates towards the midline of the cell and the centrosomes. Though a direct function is not provided, Haraguchi et al. suggested that KIF3B is necessary for proper mitosis through mutant KIF3B studies [15]. Compared to the wild-type, cells expressing the truncated form of KIF3B had an abnormal number of centrosomes, in addition to aberrant spindle fiber formation. Furthermore, mutant cells had a three-fold increase in aneuploidy. This work implicates KIF3B in the integrity of mitosis; however, its mechanism of function needs to be elucidated. Loss of function studies would be necessary for addressing this gap. 

One of the important ways in which kinesins have been shown to function during mitosis is by providing the force required to drive the centrosomes apart [16]. KIF11 is critical for this movement. KIF11 is a Kinesin-5 family member, that forms a bipolar homo-tetramer with pairs of motor heads at the opposite end of the protein complex [17]. KIF11′s unique structure allows it to span and bind two parallel microtubules and crosslink them [18]. This structure also allows for KIF11 to bind adjacent antiparallel microtubules and slide them. This results in the “push together” and “pull apart” mechanisms required to form spindle fibers [18]. KIF11 is localized along spindle microtubules and is particularly enriched near spindle poles [19]. Thus, KIF11 supports the bipolarity of the spindle fiber by creating forces directed at the centrosome between antiparallel microtubules in the spindle equator [20]. KIF11 motor activity can be regulated via phosphorylation. PTEN dephosphorylates KIF11 during mitosis. This dephosphorylation is necessary for the integrity of mitosis as hyperphosphorylation of KIF11 causes chromosome misalignment and catastrophic mitotic failure [21].

KIF15, otherwise known as HKLP2, has been shown to work cooperatively with KIF11 [22,23]. KIF15 is a plus end-directed kinesin that was first identified in the Xenopus [24]. Throughout the cell cycle, KIF15 is localized to the centrosome, however, during metaphase, it is also localized to the spindle fiber [24]. Early work by Boleti et al. showed that the motor domain of KIF15 is required for centrosome separation during spindle assembly [24]. They further speculate that KIF15 provides the pushing force that keeps spindle poles apart [24]. Later work from Vanneste et al. found that KIF15 is involved in the formation and stability of the spindle fibers [25]. Their work suggests that though KIF15 shares redundant functions with KIF11, the mechanism appears to be different between the Kifs [25]. 

Kinesin-14 family of proteins is another group of motors implicated in mitosis. In mammals, the Kinesin-14 family includes three kinesins: KIF1C, KIF2C, and KIF3C [26]. During interphase, KIFC1/HSET is sequestered to the nucleus due to its nuclear localization signal (NLS) to prevent aberrant MT crosslinking [27,28]. During mitosis, KIF1C is released to the cytoplasm where it controls MT crosslinking and sliding [27]. Like other mitotic kinesins, KIF1C localizes to the spindle fibers during mitosis. Work from Cai et al. demonstrates that KIF1C through its activities on MT controls the morphology of spindle fibers, namely its length and thickness [27].

KIF2A, a member of the Kinesin-13 family of proteins, is also associated with mitotic spindles. Generally, KIF2A localizes to the spindle poles during interphase and aids with chromosome congression [29]. Importantly, RNAi mediated knockdown of KIF2A causes significant chromosome misalignment [29]. Also, human cells deficient in KIF2A form monopolar instead of bipolar spindles during mitosis [30]. Mechanistically, KIF2A promotes spindle bipolarity through its microtubule depolymerizing activity [31]. During anaphase, KIF2A contributes to chromosome movement by disassembling microtubules at their minus ends at spindle poles [30].

Members of the Kinesin-6 family have been identified as key players in abscission, a critical stage of cytokinesis [32]. There are three members of the Kinesin-6 family: KIF20A, KIF20B, and KIF23; with KIF20B (M Phase Phosphoprotein 1, MPP1) being vertebrate-specific [32]. Work in both human and mouse cell lines showed that KIF20B first localizes to the central spindle fiber during cytokinetic furrowing and then moves to the midbody during abscission [32,33]. The knockout (KO) of KIF20B in human cycling cells causes a failure of cell division late in cytokinesis [33]. 

KIF23 (MKIp1) is another kinesin involved in cytokinesis. KIF23 is a component of centralspindlin and is required for furrow ingression, central spindle, and midbody formation [34]. MKIp1 is inactive during metaphase and becomes activated via phosphorylation when cytokinetic furrowing begins during anaphase [35]. It appears this phosphorylation is required for the proper localization and functioning of MKIp1 during cytokinesis [35]. For instance, loss of MKIp1 causes cytokinetic defects and binucleated cells [35]. 

## 3. Post-Mitotic Function of Kinesins: Microtubule Regulation 

A common theme in the action of kinesins during mitosis is the regulation of microtubules. However, the kinesins also regulate the function of microtubules in post-mitotic neurons. Microtubules (MTs) are composed of alpha and beta tubulins that are built into rod-like structures with a hollow interior [36]. Microtubules form the cytoskeleton of the neuron and help form the structure of axons and dendrites. Microtubules are dynamically altered throughout the lifespan of the neuron. These alterations can include growth/polymerization, shortening (catastrophes), and rescues (reconstitution) [37]. In axons, microtubules are organized in a parallel fashion with the polymerizing end (i.e., plus end) oriented away from the soma towards the growth cone [11]. This differs from dendrites, where microtubules are bundled with mixed orientations. In addition to their structural role, microtubules provide the “tracks” upon which cargos are trafficked throughout the cell [2]. As a result, microtubules play a fundamental role in the cellular transport of neurons.

Kinesins bind to microtubules and modulate their function (Figure 1). Eg5 (KIF11), the Xenopus Kinesin-5 family member, was found to impede dendritic spine growth and dendritic branching by inhibiting retrograde microtubule polymerization via NEK7-dependent accumulation [11]. This function was found to be independent of its motor. In effect, Eg5 acts as a microtubule stabilizer. 

## 4. Structural Plasticity

Structural plasticity is the ability to respond dynamically to the environment by altering the anatomical connectivity between neurons [38]. This change is mediated by primarily modulating the actin-cytoskeleton of axons and dendrites. Structural plasticity includes elimination, addition, and stabilization of dendritic spines, modulation of synaptic densities, growth, and retraction of axons, remodeling of dendritic branches, and even changes in cell numbers [38,39]. Kinesins play a critical role in maintaining structural plasticity by transporting necessary cellular cargos [40,41,42] and modulating the cytoskeleton through microtubules [43].

KIF3B, KIF11, KIF20B, and KIF23 act in both axons and dendrites (Table 1). KIF3B is a major regulator of dendritic and axonal architecture. For instance, Takeda et al. found that anti KIF3B antibody-mediated knockdown of KIF3B results in diminished fast axonal vesicle transport. They also found that knockdown of KIF3B, in developing neuronal cultures, produced short or completely absent neurites. Interestingly, KIF3B was found to interact with fodrin, using a yeast two-hybrid screen [44]. Fodrin is a spectrin-like structural protein that is posited to act as a linker between the KIF3B motor and transport vesicles. Moreover, it serves as an important structural component of the axon’s plasma membrane [45,46]. Interestingly, dissociated hippocampal cultures from KIF3B^+/-^ mice showed a pronounced axonal phenotype. The growth cones of KIF3B^+/-^ neurons were significantly enlarged [47]. The number of filopodia on the growth cone was decreased in KIF3B^+/-^ neurons [47]. The immature arrest of the growth cone may be the cause of the miniature neurites observed with KIF3B knockout. 

KIF3B also has a dendritic phenotype. Post-developmental knockdown of KIF3B in dissociated cortical neurons showed that the overall density of dendritic spines was increased [56]. This corresponded with an increase in both mushroom and thin spines [56]. Dendritic branching and growth are mediated by microtubule polymerization [57]. Taken together these observations suggest that KIF3B could function to negatively regulate microtubule polymerization, therefore, inhibiting the formation of dendritic processes.

KIF11 is another mitotic kinesin found to have a key role in post-mitotic neurons. Swarnkar et al. found that post-developmental knockdown of KIF11 resulted in increased dendritic branching in hippocampal neurons and total spine density [12]. This is similar to the work in superior ganglion cells, which showed increased dendritic arborization with KIF11 inhibition [58]. These studies suggest that KIF11 functions post-mitotically through the modulation of dendrites. KIF11 also has an axonal phenotype. Knockout of KIF11 causes axons to grow five times longer than controls with increased branching [20]. The phenotype appears to be caused partly by a lower propensity of the axon and newly formed branches to undergo bouts of retraction. 

KIF1C is a minus-end-directed motor that is evolutionarily conserved from simple yeasts to higher eukaryotes [26]. Apart from its function in development, KIF1C is highly expressed in the CNS throughout adulthood [48,59], indicating that it may function in post-mitotic neurons. Work from Muralidharan et al. shows that KIF1C plays a vital role in the organization of microtubules in developing axons [48]. For instance, depletion or pharmacological inhibition of KIF1C in rat embryonic hippocampal neurons resulted in shorter axons in addition to reduced secondary and tertiary branching [48]. Axonal growth was also impaired, as KIF1C depleted axons of rat spinal cord ganglion (SCG) neurons grew in spurts with periods of stalled growth [48]. This contrasts with the control group that grew steadily. The growth cones of the axons were also altered with KIF1C depletion. KIF1C depleted neurons had smaller growth cones with fewer filopodia occupied by microtubules compared to controls [48]. Furthermore, the tips of KIF1C axons were blunted and clubbed in comparison to the control, which were fan-like in nature [48]. Upon examination of fluorescently stained microtubules, KIF1C-depleted growth cones had notably more curled microtubules than in control growth cones [48]. Interestingly, this phenotype is reminiscent of stalled growth cones or growth cones of retracting axons [48]. 

Mitotic kinesin KIF15 is expressed in post-mitotic neurons [51]. Work in zebrafish has demonstrated that KIF15 plays a role in axonal development [22]. For instance, knockdown of KIF15 significantly increases the velocity of axonal growth, producing longer axons [22]. In addition, the axons had fewer branches. However, with KIF15 overexpression, the phenotype is reversed [22]. It is important to note that this post-mitotic phenotype bears some similarity to that of KIF11.

Like KIF15, KIF2A also has a pronounced axonal phenotype. Work from Maor-Nof et al. showed that KIF2A supports axonal pruning via its depolymerase activity [50]. Knockout of KIF2A causes the enhanced stability of the MTs by preventing depolymerization and degradation [50]. KIF2A also depolymerizes MTs at the growth cone edge of axons and suppresses the growth of axonal collateral branches [52]. The MT-depolymerizing activity of KIF2A appears to be enhanced by the presence of PIPKα, a key signaling enzyme [60]. Furthermore, axon elongation is suppressed by PIPKα via KIF2A [60]. Interestingly, KIF2A activity is regulated via two different phosphorylation cascades: A-type mediated by ROCK2 and B-type mediated by PAK1 and CDK5 [61]. A-type phosphorylation of KIF2A accelerates MT depolymerization, which suppresses neurite outgrowth [61]. On the other hand, the B-type phosphorylation cascade slows MT depolymerization, which promotes the extension of neuronal processes [61]. It has been demonstrated that extracellular signals appear to mediate the A and B phosphorylation of KIF2A in neurons [61]. 

KIF20B is a vertebrate-specific plus-end directed microtubule motor [32]. Work from Janisch et al. implicates KIF20B in corticogenesis [53]. Cortical neurons, from a loss of function mutant mouse against KIF20B, have short and thin apical dendrites as well as extra axonal branches [53]. These mutant neurons display a pronounced polarization defect, which appears to be partly mediated by Shootin1 [53,54]. Of the neurons that do polarize, their axons are shorter and have more branches [53]. Also, the axons of KIF20B mutant neurons tend to retract less, which may indicate that KIF20B normally functions to restrain axonal retraction [53]. Mutant neurites, on the other hand, were wider and longer with growth cone filopodia that were longer and had more microtubule invasion [53]. 

KIF23, like KIF20B, is also a plus-end-directed microtubule motor [32]. Early work from Sharp et al. [62] showed that KIF23 is expressed in post-mitotic neurons. Furthermore, they were among the first to show that KIF23 was involved in dendritic differentiation. Using an antisense oligonucleotide to knockdown KIF23 in primary neurons, Sharp et al. found dendrites failed to develop while axonal growth remained unhampered [62]. Later work from Yu et al. found that KIF23 depletion in rat sympathetic neurons, caused dendrites to grow significantly longer with thinner processes, acquiring a more axon-like phenotype [55]. The authors reasoned that KIF23 regulates the transport of minus-end microtubules into nascent dendrites [55]. In addition, KIF23 antagonizes the dynein-mediated transport of plus-end microtubules [55]. Together these functions of KIF23 contribute to the multi-polar orientation of dendritic microtubules, resulting in their short, tapered morphology [63]. However, despite the wealth of data associating kinesins with dendritic and axonal phenotypes, the mechanisms of how kinesins promote structural changes in dendrites and axons need to be elucidated.

## 5. Transport

In addition to regulating the structural plasticity through microtubule regulation, the canonical role of kinesins is the long-distance transport of cellular materials. Like trains on a track, kinesins bind to MTs and haul cellular cargos throughout the neuron. Most kinesins traffic various cargos in an anterograde fashion, towards the plus ends of microtubules. The types of cargos are diverse (i.e., RNAs, proteins, organelles, vesicles, and receptors) [1,2] and are dependent on the type and location of each kinesin in the brain [64]. 

Using immunoprecipitation and electron microscopy, KIF3B was shown to associate with membranous organelles such as endosomes [65]. These organelles were transported predominately in an anterograde fashion via fast axonal transport [44,65]. Recent work has shown that KIF3B also traffics NR2A to the synapse [47]. Experiments such as proteomic analyses of kinesin complexes have yet to be done to identify the cargos of KIF3B, 11, and other mitotic kinesins to elucidate their role in cargo transport. 

## 6. Synaptic Plasticity

Trafficking of vesicles and receptors are vital for synaptic plasticity. After first being described by Donald Hebb in 1949, synaptic plasticity has formed the cornerstone of modern neuroscience [66]. The Hebbian theory states that the output of a neuron is influenced by the firing of another neuron to with it is connected. A simplified definition of synaptic plasticity is that it is an activity-dependent modification of the strength or efficacy of synaptic transmission at preexisting synapses [67]. Kinesins largely contribute to this plasticity by trafficking vesicles and receptors that are vital for these modifications to occur. 

KIF3B and KIF11 have demonstrated involvement in synaptic plasticity. Swarnkar et al. performed an unbiased screen of 18 kinesins for their role in excitatory synaptic transmission [12]. They found that KIF11 acts as an inhibitor of synaptic transmission and is, therefore, involved in synaptic plasticity. KIF3B knockout reduced the amplitude of mini excitatory postsynaptic currents (mEPSCs), a measure of synaptic transmission indicating a functional synapse [12]. A change in the amplitude of mEPSC is usually indicative of synaptic transmission driven by a postsynaptic mechanism. This mechanism is mediated by an alteration in the number or the conductance of AMPA (α-amino-3-hydroxy-5-methylisoxazole-4-proprionic acid) receptors at the postsynaptic cell [68]. Along the same lines, Alsabban et al. showed that the ratio of NMDA/AMPA amplitudes was reduced in slices from KIF3B^+/-^ brains [47]. Furthermore, long-term potentiation (LTP), a form of synaptic plasticity resulting from the long-term increase in excitatory synaptic transmission was increased whereas long-term depression (LTD), a form of synaptic plasticity resulting from the long-term decrease in excitatory synaptic transmission was decreased in these mice. On the other hand, knockdown of KIF11 via shRNAs resulted in an increased frequency of mEPSCs suggesting a presynaptic role of KIF11 in modulating excitatory synaptic transmission [12]. This observation suggests that there is a change in the presynaptic release probability in KIF11 deficient neurons. A change in the number of readily releasable pool of vesicles or the number of synaptic sites can affect release probability [69]. Interestingly, the bath application of D-AP5, a blocker of NDMA (N-methyl-D-aspartate) receptor, abolished the increase in the frequency of mEPSCs in KIF11 knockdown neurons. This indicates that pre-synaptic NMDA receptors mediate the inhibitory effects of KIF11 on synaptic transmission [12]. Taken together, these studies suggest that mitotic kinesins play a vital role in synaptic transmission. 

It is known that in mitosis multiple kinesins coordinate their activity to maintain the fidelity of the spindle fibers [16]. However, in post-mitotic neurons, it remains unclear how multiple kinesins coordinate their pre- and post-synaptic function within the cell. Simultaneous live imaging of multiple kinesins could provide key information on this regulatory activity. Furthermore, mechanisms by which kinesins function as inhibitory constraints on synaptic transmission and plasticity have yet to be elucidated.

## 7. Higher Order Cognition

Given the role of mitotic kinesins in structural as well as synaptic plasticity, it is conceivable that their proper function would be required for cognition and might impact neuropsychiatric disorders. However, their role in cognition has been sparsely understood. Progress has been made in the case of KIF3B and its role in learning and memory. Alsabban et al. investigated the impact of KIF3B knockout (KO) on various types of learning and behavior using whole-body mutant mice [47]. They found that during the elevated plus-maze assay, KIF3B^+/-^ mice spent more time and had more entries into the open arms indicating an anxiolytic phenotype [47]. In addition, these mice exhibited impaired social interaction in the three-chamber sociability test. Compared to the control, KIF3B KO mice showed diminished spatial memory formation as well as reduced cognitive flexibility in the Barnes and Reverse Barnes tests, respectively.

Few studies have described the association of mitotic kinesins in human diseases where cognition is altered: KIF3B in Schizophrenia [47] and KIF11 in Alzheimer’s disease [70]. Studies performed in disease models would help uncover the role of mitotic kinesins in these pathological states. 

## 8. Conclusions

Consistent with their dual role, mitotic kinesins mediate the spindle and chromosome motility, flagellar and ciliary function, and vesicular and organelle transport. In mitosis, groups of kinesins provide the motile forces required to form and orient spindle fibers, both spatially and temporally. However, little is known about the mechanistic basis of their temporal regulation during mitosis. On the post-mitotic side, kinesins have variable functions (Table 1). Mitotic kinesins play a role in synaptic plasticity directly or indirectly by mediating the trafficking of receptors and synaptic proteins [4,42,47,71]. Through their ability to regulate MT dynamics, mitotic kinesins can modulate the growth of dendrites, dendritic spines, as well as axons. Despite this knowledge of mitotic kinesins in post-mitotic neurons, several questions remain: Are mitotic kinesins expressed in a cell-specific way in the brain? Are they regulated differently during development and aging? Do they function in learning and memory? Who are their interacting partners? These questions are highly pertinent to advancing knowledge regarding the post-mitotic function of mitotic kinesins.

Furthermore, one of our earlier studies [64] found the kinesin KIF5C could have different protein cargos based on where it is expressed in the brain. This would indicate that kinesins also function differently based on their location in the brain. Thus, it will be fascinating to identify cell-specific and region-specific interactors of mitotic kinesins. In addition, there could be several other kinesins that might play a critical role in mitosis. One fertile area for searching for these candidates is cancer biology. Many kinesins that are found to be associated with cancer, tend to function in mitosis [72]. Exploring mitotic kinesins and their post-mitotic functions is expected to provide novel mechanistic insights into the regulation of neuronal architecture, synaptic transmission, and plasticity. 

## Figures and Tables

**Figure 1 cells-10-00136-f001:**
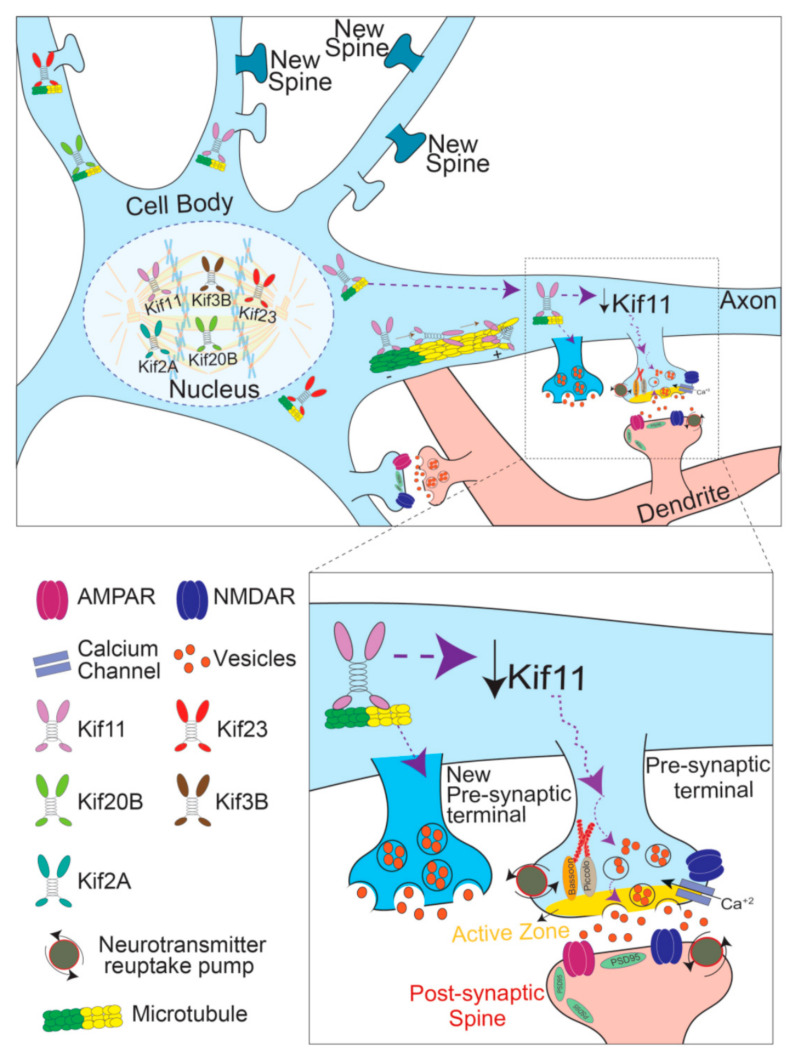
Cartoon illustration of known functions of mitotic kinesins in post-mitotic neurons. Functions of Kifs during cell division are depicted in the center surrounded by a dotted circle: KIF11/KIF2A/KIF3B/KIF20B/KIF23 have several key roles during mitosis including force generation and cargo trafficking. In the cell body: KIF11/KIF20B/KIF23 regulate microtubules by modulating their growth rate and stability. The inset shows the role of KIF11 in modulating synapse function. A reduction of KIF11 causes an increase in excitatory synaptic transmission and synapse density [12]. Dotted arrows represent unknown mechanisms.

**Table 1 cells-10-00136-t001:** Mitotic kinesins and their known functions in post-mitotic neurons

Mitotic Kinesin	KIF Family	Function in Neurons	Ref
KIF1C	kinesin-14	Sorts MT into parallel bundles and cross link MTs; promotes axonal growth and the stability of the growth cones	[27,48,49]
KIF2A	kinesin-13	MT depolymerase; promotes axonal pruning	[50]
KIF3B	kinesin-2	Traffics component of the plasma membrane and NR2A to the synapse; reduces the density of dendritic spines	[44,47]
KIF11	kinesin-5	MT stabilizer; impedes dendritic spine growth and dendritic branching; suppresses axonal growth	[11,12,20]
KIF15	kinesin-12	MT depolymerase; suppresses the growth of axonal collateral branches	[51,52]
KIF20B	kinesin-6	May bundle MTs into tight arrays; regulates neuron polarization, and axon and dendrite branching, outgrowth	[53,54]
KIF23	kinesin-6	Transports minus-end MTs into nascent dendrites	[55]

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
