# Peer review of "Double Duty: Mitotic Kinesins and Their Post-Mitotic Functions in Neurons"

_cells, 2021, doi:10.3390/cells10010136_

Round 1

Reviewer 1 Report

This revised manuscript is clearly improved in its coverage of kinesins that function in both mitosis and in neurons, but it is sad to report that the scholarship shown here has still not reached a publishable level. The paper still fails to discuss the Kinesin-14s, which are of well-established important in the formation of mammalian spindles and are now known to be important in neurons as well:

Microtubule Minus-End Binding Protein CAMSAP2 and Kinesin-14 Motor KIFC3 Control Dendritic Microtubule Organization.

Cao Y, Lipka J, Stucchi R, Burute M, Pan X, Portegies S, Tas R, Willems J, Will L, MacGillavry H, Altelaar M, Kapitein LC, Harterink M, Hoogenraad CC. Curr Biol. 2020 Mar 9;30(5):899-908.e6. doi: 10.1016/j.cub.2019.12.056. Epub 2020 Feb 20. PMID: 32084403 Free PMC article.

Here, we found that the kinesin-14 motor KIFC3 is important for organizing dendritic microtubules and to control dendrite development. ...In mammals, there are three kinesin-14 members, KIFC1, KIFC2, and KIFC3.

Mitotic Motor KIFC1 Is an Organizer of Microtubules in the Axon.

Muralidharan H, Baas PW. J Neurosci. 2019 May 15;39(20):3792-3811. doi: 10.1523/JNEUROSCI.3099-18.2019. Epub 2019 Feb 25. PMID: 30804089 Free PMC article.

KIFC1 (also called HSET or kinesin-14a) is best known as a multifunctional motor protein essential for mitosis. The present studies are the first to explore KIFC1 in terminally postmitotic neurons. ...Knowledge about KIFC1 may help researchers to devise strategies f …

The current text has been improved, even though it still sounds naïve about kinesin functions and their complexities (motors that can switch direction of motion, motors that both move on the MT lattice and also affect MT dynamics, etc).  I also think that the authors could make better use of the table they have introduced and thereby shorten their text.  With those changes and the necessary addition of comments on Kinesin-14, I expect that a publishable paper may result.

Specific Comments:

The statements about the numbers of kinesins are applicable only to mammals.  Plants and protists have added many more than the 45 on Hirokawa’s list.  Be specific about the range of biology you are discussing.

Ln 90 Neurons are only one among many cell types that can become post-mitotic.

The paragraph beginning ln 98 is poorly organized and is not a sensible way to start a discussion of mitotic kinesins.  The choice of a kinesin 2 as a mitotic kinesin is strange, because the evidence for its function in mitosis is so limited, whereas other kinesin have very well established mitotic roles. All that is available for kinesin 2 is that mitosis is abnormal when this kinesin is altered by mutation.  There is no null phenotype and no evidence about its normal role in the process. Mitotic failure could well be a neomorphic activity of the mutant allele.

Paragraph beginning ln 197. It seems a major omission to leave out a mention of actin in the regulation of dendritic spines.  It is a key example of a place where MTs do not play a dominant role.

Ln 208  The use of a neutralizing antibody is NOT a knockout.

Ln 240 shows a willingness to consider organisms other than mammals.  If this were done more extensively, this review would have more substance.  Fruit flies, nematodes, and other organisms with rich traditions of genetic analysis could expand what is said here.

Reviewer 2 Report

In this manuscript, these authors reviewed the functions of five kinesins in mitotic and post-mitotic neuron. Different kinesins participated in distinct role of spindle fiber formation. In addition, these kinesins also exhibited distinct function in neurite outgrowth, cargo transport, and synaptic plasticity. This manuscript still have several major points and should be revised before it could be accepted for publication.

Major points:

  1. In general, kinesin family proteins contain more than forty members, why these authors only selected five kinesins in this review paper?
  2. The “ROLE IN HIGHER ORDER COGNITION” section could add more information such as kinesin 11, 5b, and 3b in cognition.
  3. Too many grammar errors in this manuscript, please recheck carefully.
  4. The author should also provide a summary of these kinesins in mitotic function as figure 1.

Minor points:

  1. Line 223-225, “This corresponded with an increase in both mushroom and thin spines. It is important to note that Alsabban et al. shows a reduction in spine density, stubby spines, and filopodia(39).” This sentence should be rewritten.
  2. Line 302-304, “Work must be done to identify the cargos of KIF3B, 11, and other mitotic kinesins. For instance, proteomic screens of KIF3B, 11, cargos would help to address this gap.” This sentence should be rewritten.
  3. Line 329-331, “On the other hand, knockout of KIF11 via shRNA mediated knockdown resulted in an increased frequency of EPSCs suggesting a presynaptic effect on synaptic transmission.” This sentence should be rewritten and added reference.
  4. Line 360-361, “There is evidence to suggest that mitotic kinesins are associated with human diseases where cognition is altered such as Schizophrenia and Alzheimer’s disease”. These authors should indicate which kinesins involved in Schizophrenia and Alzheimer’s disease.

Reviewer 3 Report

This review summarizes the current knowledge of the role of kinesins in regulating neuronal structure and function. The review is, in general, well written, and there are not many reviews about this specific topic. Thus, even if the theme can appear to be of interest to a specialized audience, it is novel, and therefore I think this can legitimate the publication in "cells".

Minor suggestions

Lane 84; "kinesins are also known to destabilize microtubule". It is not clear whether this is a general role of kinesins or the M-type (see lane 81)

Lane 108-120; this paragraph describes KIF3A and KIF3B (misspelled in lane 109) as part of the kinesin-2 family; is this correct? As a non-expert, I have been sometimes confused by the organization of the 45 members in subfamilies and groups. I suggest making this more explicit and referring to the table, here and wherever applicable in the text (e.g., lane 124).

Lane 225; I suggest explicit the meaning of "Fab mediated knockdown" or considering removing it.

Lanes 288-290; it is probably worth mentioning the kinase cascades involved in A- and B-type phosphorylation.

Lane 330; also here, it is probably worth mention a few examples of organelles (or cargo carriers?) transported by KIF3B.

Lane 351-370; it difficult for a non-expert to understand the biological relevance of the described kinesins' role on sEPSCs, mEPSCs, NMDA/AMPA, and the other parameters. Probably a short comment could be of help.

Finally, I have a simple curiosity. By reading the review, I have got the impression that KIF3B appears to be involved in many roles. Is this a specific feature of this Kinesin or reflects only the limited knowledge of the role of other members?

Round 2

Reviewer 2 Report

Dear Sir:

This manuscript demonstrated the biological functions of kinesin family proteins in mitotic and post-mitotic neuron. During mitosis, different kinesins are involved in distinct functions such as microtubule regulation and centrosome division. In addition, kinesins also play a critical role in axon and dendrite formation of non-mitotic neurons. This manuscript comprehensive reviewed the kinesins’ functions. In my opinion, this manuscript could be accepted after some minor modifications.

Minor points:

  1. Line 96, “Neurons are among several cells that can exit the cell cycle” this sentence should be rewritten.
  2. Line 223, “MPP1) being vertebrate-specific. (28).” should be “MPP1) being vertebrate-specific (28).”
  3. Several grammar errors such as “anaphase(31).” Should be “anaphase (31).” Were found. Please carefully re-check.

Author Response

This manuscript is a resubmission of an earlier submission. The following is a list of the peer review reports and author responses from that submission.

Round 1

Reviewer 1 Report

This manuscript comprehensively reviewed the kinesin family proteins in mitosis and post-mitosis function in neuron. Kinesins involve in the microtubule assembly which plays a critical role in mitosis. In addition, kinesins control neuron structure, proteins transport, and memory. The manuscript needs major revisions before accepting for publication.                                                                                                                                                                                                                                                                                                                                                                                                                                                                                                                                                                                                                                                                                                                                                                                                                                                                                                                                                                                                                                                                                                                                                                                                                                                                                                                                                                                                                                                                                                                                                                                                                                                                                                                                                                               

Major point:

  1. The resolution of Figure 2 is too low. Please provide higher resolution pictures.
  2. The English written is poor and should be edited.

Minor points

  1. Several grammar errors and typos should be corrected all through the manuscripts. Please carefully rechecked.
  2. Several sentences should cite reference. Please carefully rechecked.

Reviewer 2 Report

This paper is presented as a discussion of the dual roles of some kinesins in mitosis and in neural functions. Unfortunately, the paper falls short of this goal in several ways. 1) The authors miss at least one mitotic kinesin that is well demonstrated to play a significant role in neurons, the Kinesin 6, first identified as MKLP1 and subsequently called Kif20b, which is important in aspects of anaphase B or cytokinesis and in neurite outgrowth and neuron polarization, e.g., Janisch et al, 2013 Development. 140(23): 4672–4682. doi: 10.1242/dev.093286. There may be other dual function kinesins that I didn't spot, but the point is that the review is not complete.

2) Kinesins, such as Kif21A and B are discussed at some length with respect to their roles in neural function, but no mitotic role is mentioned.  Indeed, I know no mitotic role for these kinesins, and when I did a MedLine search for Kif21 and Mitosis, I found no papers.  So what is this discussion doing in a paper whose title focuses on kinesins that play roles in both neurons and mitosis?

3) The material included in this review is very uneven. The descriptions of cell cycle and mitosis are at the level of a high school biology text and really have no place in the scientific literature.  The point of view of the paper is that mitosis and neurons are limited to mammals, and nothing is done to take advantage of the large literature that discusses these biological processes/entities in other organisms, where often more is known about molecular functions than in mammals. 

4) The description of kinesin functions in neurons is incomplete. In addition to the kinesins mentioned that are at work in both mitosis and neurons, the authors mention Kif21A and B, but they do NOT mention many other kinesins that are expressed and functional in neurons; the Kinesin 1s are poignant examples, being neuronal and the first kinesins discovered.

The text is not constructed in an economical way. Some tables that presented the motors under consideration and their functions in each biological context, followed by references, would be a succinct way to set the stage for a serious discussion of the very interesting point that they make. It might then be possible to use journal space to address fundamentally interesting points, such as whether a given motor performs the same function in these two contexts or is actually doing something different.  If the latter, how is this duality accomplished and/or regulated.

In summary, although the paper includes some interesting points, it fails to achieve the task set by its title and is not suitable for publication.

Reviewer 3 Report

In this manuscript, Joseph et al. summerise recent advances on the roles of “mitotic kinesins” in post-mitotic neurons.

The topic is of great interest and suitable for this journal. However, pitfalls are readability and accuracy. Although the authors have attempted to cover the relevant area, descriptions and citations may not be articulate.

Major points

1)Types of kinesins

In many places, kinesin molecules are introduced without explaining which types of kinesins they belong to. I recommend amending this sloppiness. This suggestion includes Figure 1 and its legend.

2) citations

Again in many places, descriptions are made without listing relevant publications (eg. lines 175-180). Sometimes only the name of the first author is mentioned (eg. lines 226, 318 and 320). These need to be corrected.

3) internal terminal kinesins: line 90

Please explain the characteristics of this type of kinesin: microtubule depolymerase

4) human diseases: line 358-361

Kinesin 5 is linked to tumor cell invasion and ciliopathies. The most recent paper on this topic is Zalenski et al. (2020, Sci Rep, 10, 13946) and related publications are cited in this paper.

Minor points

5) line 69

the unique ability to exit the cell cycle

Please add more descriptions as to why this property is unique.

6) line 116

in addition to, aberrant------ In addition to aberrant

7) lines 121 and 122

by providing the force required to move the centrosomes to the opposite sides of the nucleus

More precisely, it would be “by providing the force required to separate the duplicated centrosomes”.

8) lines 129

Eg5, the Xenopus kinesin 5 member

9) lines 133-135

Please cite papers that show phosphorylation of kinesin 5 (eg. Blangy et al., 1995)

10) lines 333

“Second”

There is no “First” in the preceding place.